# Correlation between Ultra-Wide-Field Retinal Imaging Findings and Vascular Supra-Aortic Changes in Takayasu Arteritis

**DOI:** 10.3390/jcm10214916

**Published:** 2021-10-24

**Authors:** Barthelemy Poignet, Philippe Bonnin, Julien Gaudric, Ismael Chehaibou, Mathieu Vautier, Ramin Tadayoni, Alain Gaudric, Michel Paques, Bahram Bodaghi, David Saadoun, Sophie Bonnin

**Affiliations:** 1Ophthalmology Department, Hopital Pitié Salpêtrière, Sorbonne University, 75013 Paris, France; bahram.bodaghi@aphp.fr (B.B.); sbonnin@for.paris (S.B.); 2Clinical Physiology and Fonctionnal Explorations, Hôpital Lariboisière, University of Paris, 75010 Paris, France; philippe.bonnin@aphp.fr; 3Vascular Surgery Department, Hôpital Pitié Salpêtrière, Sorbonne University, 75013 Paris, France; julien.gaudric@aphp.fr; 4Ophthalmology Department, Hôpital Lariboisière, University of Paris, 75010 Paris, France; ismael.chehaibou@aphp.fr (I.C.); ramin.tadayoni@aphp.fr (R.T.); alain.gaudric@aphp.fr (A.G.); 5Immunopathology, Immunotherapies of Autoimmunes and Inflammatory Diseases, Hôpital Pitié-Salpêtrière, Sorbonne University, 75013 Paris, France; mathieu.vautier@aphp.fr; 6Department of Internal Medicine and Clinical Immunology, Hopital Pitié-Salpêtrière, Sorbonne University, 75013 Paris, France; david.saadoun@aphp.fr; 7Ophthalmology Department Fondation Adolphe de Rothschild, 75019 Paris, France; 8Clinical Investigation Center CIC 1423, Quinze-Vingts Hospital, Sorbonne University, 75012 Paris, France; mpaques@15-20.fr

**Keywords:** Takayasu, arteritis, microaneurysm, ultra-wide-field imaging, hypoperfusion

## Abstract

(1) Background: Takayasu arteritis (TA) is a chronic inflammatory large-vessel vasculitis. Ultra-wide-field imaging allows describing the retinal lesions in these patients and correlating them with vascular supra-aortic stenosis. (2) Methods: In total, 54 eyes of 27 patients diagnosed with TA were included, and a complete ophthalmological examination was performed, including UWF color fundus photography (UWF-CFP), fluorescein angiography (UWF-FA), and computed tomography angiography measuring supra-aortic stenosis. Eleven patients underwent Doppler ultrasound imaging assessing the blood flow velocity (BFV) in the central retinal artery (CRA). (3) Results: Microaneurysms were detected in 18.5% of eyes on fundus examination, in 24.4% of eyes on UWF-CFP, and in 94.4% of eyes on UWF-FA. The number of microaneurysms significantly correlated with the presence of an ipsilateral supra-aortic stenosis (*p* = 0.026), the presence of hypertension (*p* = 0.0011), and the duration of the disease (*p* = 0.007). The number of microaneurysms per eye negatively correlated with the BFV in the CRA (r = −0.61; *p* = 0.003). (4) Conclusions: UWF-FA improved the assessment of TA-associated retinal findings. The significant correlation between the number of microaneurysms and the BFV in the CRA gives new insight to our understanding of Takayasu retinopathy. The total number of microaneurysms could be used as an interesting prognostic factor for TA.

## 1. Introduction

Takayasu arteritis (TA) is a chronic inflammatory large-vessel vasculitis, predominantly affecting the aorta and its main branches [1]. Vessel inflammation leads to wall thickening, fibrosis, stenosis, and thrombus formation [2]. TA primarily affects women, and its incidence varies depending on the geographic area with a predominance in South East Asia [3].

Several related ocular manifestations have been described, such as ocular ischemic syndrome and arterial retinal occlusion. As early as in 1977, Uyama and Asayama [4] have defined a retinopathy grading based on the presence of microaneurysms or arterio-venous shunts and, in the most advanced cases, ischemic complications. Most lesions have been described at the posterior pole or in the mid-periphery, depending on the imaging device used.

Advances in imaging techniques, and especially the possibility to perform ultra-wide-field (UWF) imaging, now allow a more thorough assessment of these peripheral retinal lesions, giving the unique opportunity to assess more broadly and precisely the consequences of low ocular perfusion potentially induced by a supra-aortic stenosis.

Combining an UWF ophthalmologic examination and a detailed systemic assessment, including supra-aortic stenosis and the blood flow velocity (BFV) in the central retinal arteries, allows correlating the retinal findings with the presence of a supra-aortic vascular involvement.

As these patients are young, in the absence of any other ocular disease or media opacity, this retinal disease could also be a very interesting model of hypoperfusion-induced retinopathy.

The aims of this study were to assess the retinal manifestations of TA on UWF color fundus photography (UWF-CFP) and fluorescein angiography (UWF-FA), and to investigate the correlation between the severity of retinopathy and the presence of vascular supra-aortic changes.

## 2. Materials and Methods

In this retrospective study, the records of consecutive patients with TA referred to our department between November 2018 and November 2019 were reviewed. This study was approved by the Ethics Committee of the French Society of Ophthalmology (IRB 00008855 Société Française d’Ophtalmologie) and adhered to the tenets of the Declaration of Helsinki. Informed consent was obtained from all patients.

Inclusion criteria were patients with TA, meeting the American College of Rheumatology classification and/or Ishikawa criteria modified by Sharma criteria [5], aged 18 years or older, and imaged with sufficient quality UWF-CFP and UWF-FA. Exclusion criteria were any contraindication to fluorescein injection, pregnancy, poor-quality images due to media opacity, or the presence of other retinal diseases (including high myopia and diabetic retinopathy).

The following data were assessed: age, gender, medical and surgical history, mean TA duration, clinical presentation (acute or progressive), past and current treatments received by the patient (surgery, immunosuppressive treatment, systemic corticosteroid therapy, or simple medical observation), as well as any history of hypertension and its treatment.

The ophthalmological examination included the measurement of the visual acuity (VA) and intraocular pressure, a slit-lamp examination (rubeosis), and a fundus examination.

### 2.1. Retinal Image Acquisition

According to the current practice, the studied eyes were dilated with tropicamide 1%.

UWF images were obtained using the Optos California v4.3 (Optos, PLC, Dunfermline, Scotland). After intravenous injection of fluorescein, UWF-FA images were captured in the early (45 s), intermediate (2 min), and late (5 min) phases of FA. For both UWF-CFP and UWF-FA, the eyes were steered peripherally toward 4 directions (nasally, temporally, superiorly, and inferiorly), in order to acquire 5 pictures in total.

### 2.2. Image Grading

Each eye was graded according to the Uyama and Asayama classification, i.e., 1 = simple vein dilations, 2 = microaneurysms, 3 = arteriovenous shunts, 4 = ischemic complications.

On both CFP and FA images, the number of microaneurysms was graded in the four peripheral retinal quadrants and at the posterior pole (1 when <5 microaneurysms were present, 2 when 5–30 microaneurysms were present, 3 when >30 microaneurysms were present). The number of microaneurysms was also determined in each eye by 2 graders (BP, SB), using all FA images available.

The presence of hypertensive retinopathy was assessed based on prior published classifications [6,7]. Graders checked for signs of acute hypertension (retinal hemorrhages, hard exudates and papilledema) and chronic arteriosclerosis (cotton wool spots, vascular narrowing, and arteriovenous nicking).

### 2.3. Supra-Aortic Stenosis Assessment

In order to assess the vascular lesions of the supra-aortic trunks, as usually performed in clinical practice, all patients underwent Computed Tomography Angiography (CTA): the severity of stenosis in the common carotid artery, in the internal carotid artery, and in the brachiocephalic trunk was assessed and expressed as a percentage between the residual lumen diameter and the estimated vessel diameter. All stenoses >30% were considered significant. The clinical type of thoracic aortic involvement was graded from I to V according to the angiographic classification [8].

### 2.4. Ultrasound Imaging

To better understand the pathophysiology of these retinal lesions, 11 consecutive patients underwent ultrasound imaging to measure the BFV in the retrobulbar arteries.

Color-coded and pulsed Doppler ultrasound imaging was performed using an ACUSON S2000 ultrasound system (Siemens, Erlangen, Germany) equipped with a linear transducer type 18L6HD (18 Mhz). The BFV was directly measured in the lumen of each studied retrobulbar artery. The transducer was applied to closed eyelids without exerting any pressure on the bulb. Two-dimensional ultrasound imaging allowed distinguishing anatomical structures, in particular the eyeball and the optic nerve that were used as landmarks. Then, the color-Doppler mode was activated to identify the circulating lumen of each studied artery and its direction. Subsequently, the retrobulbar vessels were observed on the screen based on their color-coded flow: central retinal artery (CRA), temporal posterior ciliary artery (TPCA), and ophthalmic artery (OA) as previously described [9]. A pulsed Doppler sample was set at 1 mm to only record the waveform of the blood flow in the studied artery. It was placed in the CRA, where it passed through the optic nerve, about 0.5 mm upstream its entry into the sclera, and then, a spectral analysis of the Doppler BFV waveforms was recorded. Then, the Doppler sample was moved into the lumen of the long TPCA, about 0.5–1 cm upstream its entry into the sclera, and then into the lumen of the OA, where it passed through the optic nerve. From each Doppler velocity waveform, the peak systolic (syst-), end-diastolic (diast-), and spatial-averaged-time-averaged mean (mean-) BFVs were calculated in the three studied arteries. The right eye was examined first and then the left eye, and the investigator (PB) was blinded to the severity of the arterial lesions affecting the supra-aortic trunks.

### 2.5. Statistics

The numbers of microaneurysms were compared using Student’s t-test. Correlations between the number of microaneurysms and the quantitative variables (i.e., age or disease duration) were assessed using linear regressions. Correlations between the number of microaneurysms and the nominal qualitative variables (i.e., clinical form, presence of a supra-aortic stenosis, immunosuppressive therapy, or antihypertensive therapy) were assessed using a variance analysis (ANOVA). Correlations between the mean BFV recorded either in the CRA or in the OA and the number of microaneurysms were calculated using the Spearman correlation. A *p* value <0.05 was considered significant. All statistical analyses were performed using SAS v 9.4 TS Level 1 M6 (Copyright (c) 2002–2012 by SAS Institute Inc., Cary, NC, USA. All Rights Reserved) and GraphPad Prism 9.0 (GraphPad Software, Inc., San Diego, CA, USA).

## 3. Results

Fifty-four eyes of 27 patients were included. Patients’ characteristics are presented in Table 1. Among these patients, 92.6% (25 patients) were women. Patients’ mean age was 47 ± 13.1 years. The mean duration of TA was 10.4 ± 9.4 years. Fourteen patients (66.7%) patients were treated with at least one immunosuppressive treatment associated with systemic corticotherapy: 29.6% (eight patients) were treated with methotrexate, 11.1% (three patients) were treated with aziathioprine, and 11.1% (three patients) were treated with mycophenolate mofetil. In addition, 14.8% (four patients) of patients received a biotherapy (two with infliximab, one with adalimumab, and one with tocilizumab). In addition, 29.6% of patients had hypertension and received appropriate medication. The mean VA was 20/20 (0.018 ± 0.084 logMAR).

### 3.1. Analysis of UWF Images

The stages of Takayasu retinopathy (TR) based on UWF-CFP and UWF-FA images are reported in Table 2. Microaneurysms were visible in 18.5% of eyes on fundus examination alone.This percentage increased to 24.4% of eyes on UWF-CFP and to 94.4% of eyes on UWF-FA (Figure 1). Microaneurysms were predominantly detected in the peripheral retina and mainly located around the veins. On UWF-FA, 22% of eyes had microaneurysms visible at the posterior pole. The number of microaneurysms was significantly higher in the temporal quadrant of the peripheral retina than in the posterior pole (*p* < 0.001). Arteriovenous and veno-venous shunts were detected in 22.2% of eyes on UWF-FA (Figure 2). One patient (3.7%) presented with signs of former branch retinal artery occlusion associated with diffuse non-perfusion areas, without sign of iris rubeosis or neovascular glaucoma. The mean number of microaneurysms per eye was 113 ± 175. The intergrader reliability of microaneurysms count was excellent with a mean intraclass correlation coefficient (ICC) value of 0.977 (95% CI, 0.961–0.987) (Table 2).

### 3.2. Correlation between the Number of Microaneurysms and the Presence of Systemic Factors

The grading of microaneurysms (1–3) significantly correlated with the intake of antihypertensive treatment (*p* < 0.0001) and the disease duration (*p* = 0.092) but also with the clinical form (*p* = 0.0261) and with the type of treatment (*p* = 0.0175). No significant correlation was found between the grading of microaneurysms (1–3) and the presence of a supra-aortic stenosis. However, the total number of microaneurysms significantly correlated with the presence of a stenosis (*p* = 0.026), the presence of hypertension (*p* = 0.0011), and the disease duration (*p* = 0.0068).

### 3.3. Correlation between the Number of Microaneurysms and the BFV Measured in the OA and CRA

Ultrasound imaging allowed measuring the BFV in the CRA, which was particularly low in patients with the highest number of microaneurysms (Figure 1). Eleven patients underwent ultrasound imaging, and data were analyzable for 21 eyes. The mean BFV in the CRA significantly and negatively correlated with the total number of microaneurysms per eye (r = −0.61; *p* = 0.003) (Figure 3). No significant correlation was found between the mean BFV in the OA and the total number of microaneurysms (r = −0.12, *p* = 0.61).

## 4. Discussion

This study showed that UWF-FA allowed better detecting retinal microvascular lesions related to TA compared to UWF-CFP. Indeed, microaneurysms were detected in only 24.4% of eyes on UWF-CFP, while on UWF-FA, microaneurysms were detected in 22% of cases at the posterior pole and in 94.4% of cases when all the periphery of the retina was included. The number of microaneurysms was significantly higher in the peripheral retina than in the posterior pole (*p* < 0.001). These findings are consistent with the assumption that TR is probably underdiagnosed [10] and showed the benefit of UWF imaging.

The strict analysis of the posterior pole in our cases found microaneurysms in only 22% of eyes, which is a value that is close to previous results. For example, Chotard et al. [11] have detected microaneurysms in five out of 26 eyes using OCTA. Microaneurysms were detected in up to 94.4% of cases when the peripheral fields were assessed, showing the value of UWF imaging to detect ocular manifestations of TA, which appeared to be more frequent than usually assumed [12].

A very small number of patients in our series complained of ocular symptoms, which supports the need for a systematic screening strategy, including UWF-FA, since TR has been shown to be a significant risk factor for TA-related vascular complications [13]. Indeed, the presence of retinal lesions has been associated with higher mortality and cumulative incidence of vascular complications in TA patients, using multivariate models [13]. A prognostic score has been developed based on the progressive course of the disease, the thoracic aorta involvement, and the presence of retinopathy [14]. Given the widespread availability of UWF retinal imaging systems in specialized retinal practice, patients diagnosed with TA should systematically undergo UWF-FA to better assess the presence of retinopathy and thus improve the evaluation of TA prognosis, at the time of diagnosis. As previous studies [13] identified retinopathy as a risk factor for vascular complications using standard ophthalmological exam, further studies should be conducted to confirm these findings using UWF imaging.

The pathophysiology of TR remains controversial, and its identification could improve the understanding of the prognostic factors for TA. Two hypotheses could explain the occurrence of retinopathy: (i) the supra-aortic stenosis could induce retinal hypoperfusion and subsequent microaneurysm development; and (ii) retinal microvascular inflammation could explain the presence of microaneurysms. In favor of the hypothesis of a hypoperfusion, we found a significant correlation between the number of microaneurysms and the mean BFV in the CRA (r = −0.61, *p* = 0.003) (Figure 3). The reduced blood flow in TA triggers vascular insufficiency of the extremities and possible cerebral and ocular ischemia. The predominant location of retinal microaneurysms around the venules supports the hypothesis of chronic retinal ischemia. Vascular stenosis could induce a chronic hypoperfusion, leading to increased intraocular levels of Vascular Endothelial Growth Factor (VEGF). Intraocular injections of VEGF in normal eyes of cynomolgus monkeys have been shown to induce the formation of microaneurysms and preretinal neovascularization originating only from superficial veins and venules throughout the peripheral retina [15], explaining the pathophysiology of the ocular ischemic syndrome, the characteristics of which are very close to TR findings. Interestingly, this study also showed a predominant involvement of the peripheral retina compared to the posterior pole. However, no significant correlation was found between the number of microaneurysms and the mean BFV in the OA (r = −0.12, *p* = 0.61). Nevertheless, the OA supplies not only the CRA but also all intra-orbital organs such as the muscles and lacrimal glands, as well as the skin at the nose root. Therefore, changes in BFV in the OA cannot reflect changes in the retinal vascular network, while BFV changes in the CRA can. Other authors have found variable results regarding ocular findings and blood flow in TA patients [12,16]. Matos et al. [16] have shown a statistically significant correlation between the peak systolic velocity in the OA and Heidelberg retinal angiography findings (*p* = 0.006), using the Uyama and Asayama classification. Thus, the decrease in peak systolic velocity in the OA could be interpreted as the expression of the presence of a significant stenosis upstream in the carotid arteries, generating a downstream low ocular supply. More recently, Esen et al. [12] have assessed the ocular blood flow in the OA and CRA using color Doppler ultrasound in 65 patients with TA, but only 6% of patients underwent FA to confirm the diagnosis of TR: the ocular blood flow was reduced in these four patients with retinopathy, but no statistical analysis was performed due to the relatively small number of patients in the retinopathy group.

While retinopathy improvement has been described after carotid reperfusion, from an increased VA [17] to the complete regression of microaneurysms [10], this reperfusion effect is variable because of the reorganization and vascular shunt developing progressively in case of supra-aortic occlusion. The same reason could explain the absence of strong correlation between the BFV in the OA and the total number of microaneurysms. We assumed that the flow in the CRA could be more affected by the vascular reorganization, leading to blood flow decrease and more retinal abnormalities because it is an end-terminal vascular network without any pre-existing arterial collateral network. Microaneurysm formation was more pronounced in the peripheral retina than at the posterior pole probably because it is located further away from the origin of the CRA.

UWF-FA images allowed detecting a higher number of retinal lesions than classic images, increasing the possibility to assess the correlation between the ocular findings and the blood flow. Our study supports the hypothesis that the number of microaneurysms seen on the UWF-FA images could be used as an interesting marker of retinal hypoperfusion severity, since it significantly correlated with the blood flow in the CRA.

On the other hand, it is well known that other clinical manifestations of TA are caused by inflammation of all the layers of the arterial wall. The narrowing of the lumen caused by intimal proliferation and fibrosis has been described on histological examination [18]. The main pathological features are granulomatous inflammatory panarteritis with intimal proliferation and disruption of the elastic lamina. Although the pathogenesis is unclear, acute vascular inflammation is observed. This could be due to the activation of the MICA/NKG2D pathway in response to an unknown stimulus, resulting in the activation of T and NK cells recruited to the vascular wall. The secondary production of pro-inflammatory cytokines and interferon-γ by Th1 lymphocytes induces the activation of macrophages, the release of VEGF and PDGF, and intimal proliferation [18]. This inflammation results in a reduction in large vessel lumen, mainly in the subclavian arteries and aortic arch branches, but it can occur in any segment [19].

The persistence of a decreased retinal blood flow in the CRA could suggest an alteration of small vessel perfusion caused by a chronic hypoperfusion but also by microvascular remodeling, explaining the persistence of microaneurysms and TR lesions in the absence of any significant supra-aortic stenosis. The presence of a high number of bilateral peripheral microaneurysms despite the presence of bilateral supra-aortic stenosis also supports the presence of an additional factor besides hypoperfusion.

We found a significant relationship between the presence of a supra-aortic arterial stenosis and the number of microaneurysms. Although these findings suggest that the number of microaneurysms could be a marker of TA severity, no correlation between the number of microaneurysms and the percentage of supra-aortic arterial stenosis was found. It could also be explained by the multiple surgical procedures performed to preserve the carotid blood supply and blood pressures toward the central nervous system and the eyes as well as the presence of vascular shunts in more than 20% of cases.

Vascular inflammation could induce both a chronic hypoperfusion and an alteration of the retinal microcirculation. Retinal capillary microaneurysms can be found in various retinal vascular diseases, especially in diabetic retinopathy, but can also be associated with hypertensive retinopathy. In our series, 29.6% of patients received one antihypertensive treatment, and a significant correlation between the intake of antihypertensive treatment and the number of microaneurysms was found, suggesting that the presence of hypertension could worsen the ischemic ocular manifestations of TA.

To improve the pathophysiological understanding of the ocular manifestations of TA, further studies using other innovative imaging techniques, such as laser Doppler imaging [20] that has already shown its potential to analyze blood flow changes and ocular hemodynamics in healthy and diseased eyes [21], are needed. The quantitative analysis of the BFV in the CRA could allow distinguishing the part of hypoperfusion explaining the ocular manifestations of TA.

Rather than investigating all the ophthalmological manifestations of TA, we chose to focus on microaneurysms and their correlation with prognostic factors of the disease. However, the presence of microaneurysms lacks specificity, since up to 41% of normal eyes may present some microaneurysms in the peripheral retina [22], but the especially high prevalence of microaneurysms in these patients and the possibility to count these lesions give some value to these findings. UWF imaging also has some limiting biases, such as a poor visualization of the posterior pole and a limited visibility in the far upper and lower retinal periphery because of the eyelids.

Other limitations of this study include its retrospective design and the relatively small number of patients included. However, considering the low incidence of TA in Europe, our cohort still represents a large sample of patients.

## 5. Conclusions

UWF-FA improved the detection of TR in this cohort, and the number of microaneurysms in the peripheral retina could be used as a new marker of retinal involvement severity in TA patients.

## Figures and Tables

**Figure 1 jcm-10-04916-f001:**
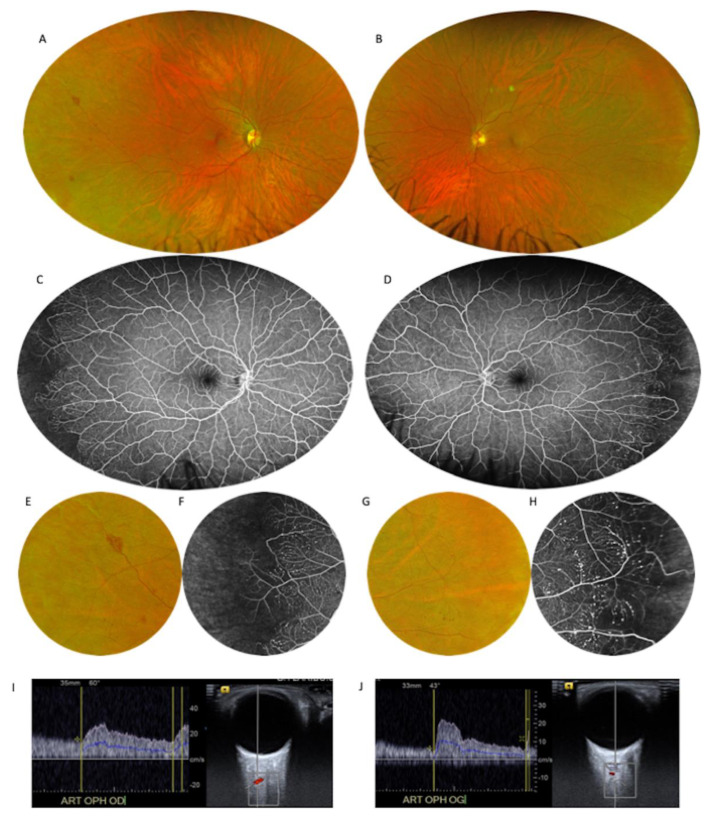
Forty-six-year-old patient with Takayasu arteritis with a history of carotid-carotid bypass and right internal carotid occlusion. (**A**,**B**): Ultra-wide-field color fundus photography (UWF-CFP) showing several punctiform hemorrhages in the temporal periphery. (**C**,**D**): Intermediate-phase ultra-wide-field fluorescein angiography (UWF-FA) revealing numerous peripheral microaneurysms in both eyes. (**E**,**F**): Close-up images of the right eye peripheral retina showing retinal hemorrhages and microaneurysms around a terminal vein. The high number of microaneurysms is associated with non-perfusion areas in the far periphery. (**G**,**H**): Close-up images of the left eye peripheral retina with visible peripheral microaneurysms. Capillary non-perfusion areas and a high number of microaneurysms are seen in the UWF-FA close-up. Doppler ultrasound of the right ophthalmic artery (**I**) and the left ophthalmic artery (**J**) shows a significant reduction in blood flow velocity.

**Figure 2 jcm-10-04916-f002:**
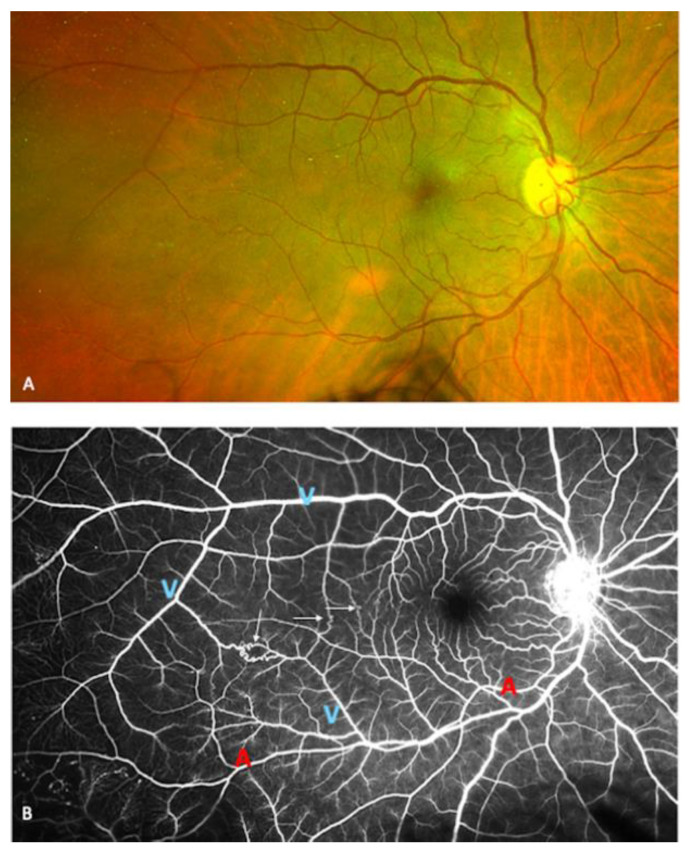
Seventy-one-year-old patient with Takayasu arteritis with a history of previous aortic and carotid thrombosis. (**A**): Ultra-wide-field color fundus photography (UWF-CFP) showing a mild venous dilation without hemorrhage or sign of vein occlusion. (**B**): Multiple arterio- and veno-venous shunts are visible on ultra-wide-field fluorescein angiography (yellow arrows). Arteries are marked with the letter “A” and veins are marked with the letter “V”. A peripheral venous dilation is visible as well as large areas without capillaries.

**Figure 3 jcm-10-04916-f003:**
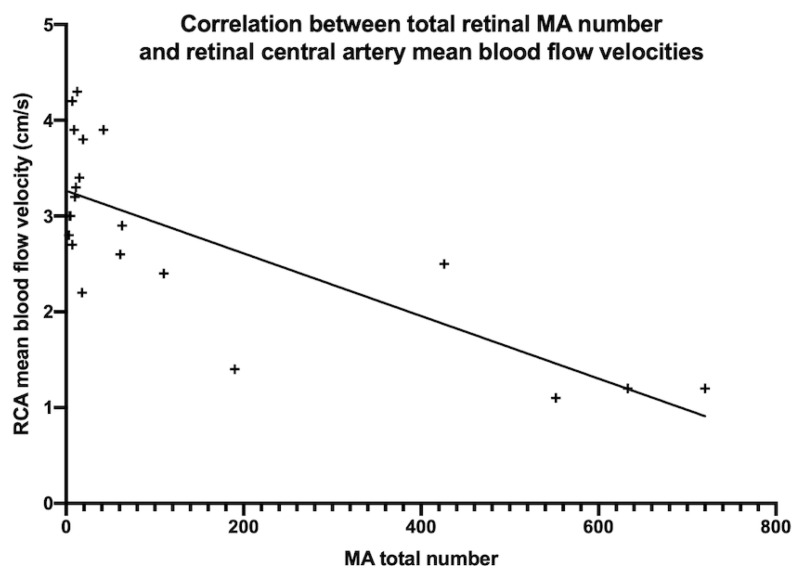
Correlation between the total number of microaneurysms and the mean blood flow velocity in the retinal central artery (RCA) using the Spearman correlation (r = −0.61, *p* = 0.0031).

**Table 1 jcm-10-04916-t001:** Patients’ characteristics.

	Total (*n* = 27)
**Gender, male (%)**	2 (7.4)
**Mean age, years (range)**	47.3 (25–84)
**Thoracic aortic involvement (%)**	
I	8 (29.6)
IIa	1 (3.7)
IIb	5 (18.5)
V	13 (48.1)
**Disease duration, years (range)**	10.4 (1–33)
**Hypertension (%)**	8 (29.6)
**Immunosuppressive therapy (%)**	
No IST	5 (18.5)
Corticosteroids alone	4 (14.8)
Methotrexate	8 (29.6)
Mycophenolate Mofetil	3 (11.1)
Azathioprine	3 (11.1)
Biotherapy	4 (14.8)
**Clinical presentation (%)**	
Acute	7 (25.9)
Progressive	20 (74.1)

IST: immunosuppressive therapy.

**Table 2 jcm-10-04916-t002:** Ophthalmological findings.

	Total (*n* = 54)
**Takayasu retinopathy stage (%), according to Uyama and Asayama classification**	
Stage 1	4 (7.41)
Stage 2	38 (70.4)
Stage 3	12 (22.2)
Stage 4	0
**Microaneurisms detection** (%)	
Clinical fundus examination	10 (18.5)
UWF Fundus photographies	13 (24.1)
UWF Fluorescein angiography	51 (94.4)
**Total number of microaneurysms per eye on UWF-FA** (**±SD)**	113 (175)
**Microaneurysms location on UWF-FA (%)**	
**Superior retinal quadrant**	
**No microaneurysm**	25 (46.3)
**<5 microaneurysms**	10 (18.5)
**5–30 microaneurysms**	15 (27.8)
**>30 microaneurysms**	4 (7.4)
**Inferior retinal quadrant**	
**No microaneurysm**	24 (44.4)
**<5 microaneurysms**	10 (18.5)
**5–30 microaneurysms**	13 (24.1)
**>30 microaneurysms**	7 (13.0)
**Temporal retinal quadrant**	10 (18.5)
**No microaneurysm**	10 (18.5)
**<5 microaneurysms**	23 (42.6)
**5–30 microaneurysms**	11 (20.4)
**>30 microaneurysms**	10 (18.5)
**Nasal retinal quadrant**	
**No microaneurysm**	19 (35.2)
**<5 microaneurysms**	11 (20.4)
5–30 microaneurysms	15 (27.8)
>30 microaneurysms	9 (16.7)

UWF = ultra-wide field; FA = fluorescein angiography.

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
