# Peer review of "Correlation between Ultra-Wide-Field Retinal Imaging Findings and Vascular Supra-Aortic Changes in Takayasu Arteritis"

_jcm, 2021, doi:10.3390/jcm10214916_

Round 1
Reviewer 1 Report
This is a retrospective study on ultra-wide-field (UWF) color fundus photography (UWF-CFP), fluorescein angiography (UWF-FA), and computed tomography angiography (CTA) findings in patients with Takayasu arteritis. These modalities seem to improve detection of the retinopathy related to Takayasu. Presence of ipsilateral supra-aortic stenosis on CTA correlated with the number of microaneurysms on UWF-FA in this study. I would like to congratulate the authors for this well conducted study with a detailed ophthalmologic evaluation.
Author Response
Thank you so much for your comments. We think ultra-wide-field imaging will greatly improve Takayasu arteritis retinal manifestations detection.
Once again thank you for reviewing our work.
Reviewer 2 Report
Congratulations on this very interesting study. I do have one comment I hope you can elaborate on. You mention that the presence of retinopathy has historically been prognostic of disease course and severity for TA. However, this was retinopathy that was detectable with dilated fundus exam. The wide field angiography allows you to detect far more retinopathy than would have been detected with DFE, and so it is not clear that detecting such mild disease will still be prognostic. Can you please comment on this in your conclusions?
Author Response
Thank you for your careful reading.
As you rightfully point out, Takayasu retinopathy was previously diagnosed using regular dilated fundus and standard field angiography. According to Comarmond et al. (2017), the retinopathy defined by microaneurysm formation was identified as a risk factor for vascular complications.
By improving this retinopathy's detection and looking for peripheral microaneurysms, this association should be confirmed.
Since the number of microaneurysms was correlated with the supra aortic vascular changes, detecting patients with high number of MA could indicate a potentially more severe vascular manifestation of this arteritis.
Moreover microaneurysms grading and total count is associated with disease duration, hypertension and even progressive clinical form, which are considered to have a worse prognosis. We could also speculate that detecting earlier manifestations could decrease the number of severe ocular complications.
However to clarify this very interesting comment, we added the following sentence in our manuscript :
Page 8, Line 238 "As previous studies (Comarmond et al.) identified retinopathy as a risk factor for vascular complications using standard ophthalmological exam, further studies should be conducted to confirm this findings using UWF imaging."